# Burnout and Engagement Dimensions in the Reception System of Illegal Immigration in the Mediterranean Sea. A Qualitative Study on a Sample of Italian Practitioners

**DOI:** 10.3390/ijerph18073726

**Published:** 2021-04-02

**Authors:** Marcello Nonnis, Mirian Agus, Monica Piera Pirrone, Stefania Cuccu, Maria Luisa Pedditzi, Claudio Giovanni Cortese

**Affiliations:** 1Department of Pedagogy, Psychology, Philosophy, University of Cagliari, 09123 Cagliari, Italy; mirian.agus@unica.it (M.A.); cuccustefania@gmail.com (S.C.); pedditzi@unica.it (M.L.P.); 2Psicologi per i Popoli Sardegna, 09128 Cagliari, Italy; m.pirrone@outlook.com; 3Department of Psychology, University of Turin, 10124 Turin, Italy; claudio.cortese@unito.it

**Keywords:** illegal immigration, JD-R model, burnout, engagement, reception system, qualitative study, NGO, volunteering, Mediterranean Sea

## Abstract

The present study describes the semantic nature of burnout and engagement in the operators involved in the management of illegal immigration. Semi-structured interviews were conducted on a sample of Italian practitioners (*n* = 62) of the two levels of the reception system considered: (1) rescue and first aid and (2) reception and integration. Within the framework of the job demands–resources model (JD-R), the interviews deepened the analysis of the positive and negative dimensions of burnout and engagement: exhaustion versus energy, relational deterioration versus relational involvement, professional inefficacy versus professional efficacy and disillusion versus trust. The interviews were analysed using the T-Lab software, through a cluster analysis (bisecting K-means algorithm), which emphasised noteworthy themes. The results show that, in the vast majority of the dimensions considered (for both levels of reception), the same dimensions of engagement of the operators (energy, relational involvement, professional efficacy and trust) are able to lead them into a condition of burnout, with experiences, conversely, of exhaustion, relational deterioration, professional inefficacy and disillusion. These findings expand the knowledge on burnout and engagement in practitioners of illegal immigration, a context characterised by the value of help and welcome.

## 1. Introduction

Illegal immigration in the geographical area around the Mediterranean Sea is an intense and epoch-making phenomenon [1,2]. Its main causes are the gap between the living conditions of disadvantaged countries on the African continent and in the Middle East compared with those of European countries, scenarios of war (civil or between neighbouring countries) and conflicts between populations of different religions. This flow has led to a real traffic of human beings who follow illegal paths, mainly by sea. In the first six months of 2020, out of about 54,000 arrivals in Europe, 23,720 migrants landed in Italy. Overall, this phenomenon puts a strain on the reception systems of Italy and other European countries around the Mediterranean Sea [3,4].

The recent COVID-19 pandemic has led to a general closure of the borders of countries. For this reason, the number of asylum applications in the first half of 2020 in Europe decreased by approximately 34% compared with the previous year [2]. However, the conditions that determine and fuel illegal migration flows have not changed, and may even be exacerbated by the economic and social consequences of the pandemic [3].

The European Union (EU) has established a community policy on asylum, aimed at offering an appropriate status to any non-EU citizen in need of international protection and requires member states to adopt rescue programmes (e.g., Sofia and Themis) and reception measures (e.g., hotspots) to ensure an adequate quality of life, health and subsistence for asylum seekers in accommodation centres [5].

Regarding Italy, the reception system is organised in two levels [6]. The first involves rescue—mostly from the sea-relief (in hotspots located in the south and islands of the country [3]) and first aid. Subsequently, the initial procedures of medical and/or psychological support, health screening, photo-signalling and identification of migrants take place within Extraordinary Reception Centres (in Italian: Centri di Accoglienza Straordinaria, CAS). In addition, CASs initiate the process of examining asylum applications and assess any vulnerable situations requiring special assistance measures. The personnel involved in this level mainly belong to law enforcement (financial police, coast guard and Frontex European border police), health sector (doctors, nurses and psychologists) and voluntary and humanitarian organisations (e.g., Caritas, Red Cross, UNHCR). This level is designed to host migrants for a relatively limited period (a few weeks or a few months at most).

The second level, called the Reception and Integration System (in Italian: Sistema di Accoglienza e Integrazione, SAI), provides for the reception of holders of international protection, and of asylum seekers. The SAI offers two levels of service. The first is intended for asylum seekers and provides a wide-ranging welcome, with health, social and psychological assistance; linguistic–cultural mediation; Italian language literacy courses; and legal services. The second level is intended for holders of international protection and is aimed at integration, job orientation and job training. These services are carried out through a dense and complex network between the SAI, social and health services, local authorities, security services, job centres, schools, voluntary organisations and social cooperatives. This level of reception implies residence times for migrants on the order of months and in some cases years [6].

In forced and illegal immigration scenarios, the multiple actors involved in helping are at risk of work-related distress, and burnout could play a major role. Burnout is an occupational syndrome that affects the service sector and specifically the social and health care contexts [7]. Its first definitions were focussed on relational aspects, especially in the context of helping professions. According to Edelwich and Brodsky [8], burnout implies the progressive loss of energy, idealism and purposefulness. It involves emotional exhaustion, the development of a negative self-concept, dissatisfaction and failure, accompanied by a feeling of helplessness [9]. Moreover, it can be considered a particular response to stress, a strenuous coping strategy that negatively affects the motivation, performance and well-being of helping professions [10]. Maslach [11,12] defined burnout as a syndrome characterised by emotional exhaustion, depersonalisation and reduced professional efficacy: a response to the chronic emotional strain produced by working with people who suffer. Subsequently, Maslach and Leiter [13,14] proposed a conceptualisation of it as a more general organisational syndrome, typical of services.

In the last twenty years, job burnout has been framed within job distress, in the job demands–resources (JD-R) model [15,16,17]. According to this perspective, job demands and job resources of different kinds (physical, psychological, social and organisational) are present in working contexts. In job burnout, in general, excessive demands are predictive of exhaustion, while inadequate resources lead to disengagement. On the other hand, adequate resources and sustainable demands generate work engagement and organisational well-being [18,19,20].

Some authors have taken up the original notion of burnout as a syndrome of helping professions [21,22,23]. In particular, Santinello et al. [24] and subsequently other authors (e.g., [25,26,27]) have proposed considering, in addition to the already mentioned dimensions of exhaustion, depersonalisation (or cynicism) and professional inefficacy, the dimension of disillusion. Indeed, expectations and motivations play an existential role for professionals with a vocation to help, which refers to their role in society and their sense of life. The theme of disillusion as an outcome of the comparison/clash between these ideals and the reality of helping work had already been proposed by Edelwich and Brodsky [8], although they had not codified it into a measurable dimension. From this perspective, burnout can also lead to the attrition and destruction of the vocational ideals and motivation to help of these workers.

Studies on burnout of practitioners working in the field of illegal migration are scarce. Among these, Apostolidou [28] and Graffin [29] conducted qualitative studies using semi-structured interviews with small samples of carers of refugees and asylum seekers (respectively in England and Ireland). Their research showed that professionals were emotionally exhausted and affected by their patients’ traumas, to the point of feeling ineffective and cynical. However, Apostolidou also pointed out that engagement in helping immigrants allowed workers to develop social engagement, a sense of efficacy, social utility and personal fulfilment. All these factors are protective against burnout.

Posselt and colleagues [30] conducted a quantitative–qualitative study with 50 practitioners working with refugees and asylum seekers in Australia. For these operators, the most significant sources of burnout were government immigration policies and bureaucratic procedures. On the other hand, sense-making in their work, also facilitated by their coordinators, played a protective role against burnout.

With regard to volunteers, Simsa et al. [31] conducted a qualitative study in Austria on 42 operators involved in the Mediterranean crisis in 2016. The authors showed that the volunteers were often called upon to compensate for the shortcomings of governmental apparatuses, and to give themselves their own organisation. The main sources of stress and burnout of volunteers were: the physical and emotional strain due to the severe suffering of the recipients; the intense and prolonged commitment and lack of rest; and the need to organise themselves in the absence of guidelines and coordination from the relevant state bodies.

Nonnis and colleagues [32] conducted a quantitative–qualitative study on a sample of Italian operators (both volunteers and professionals), representative of the Italian system of management of illegal immigration. The results of the qualitative part of the study (conducted with semi-structured interviews, *n* = 108) confirmed the findings of previous research regarding the sources of stress and burnout among staff members and highlighted difficulties due to a lack of logistical and organisational resources, cultural differences and language barriers. Finally, they showed that personal involvement and sharing positive results in helping immigrants are some of the most important protective factors against burnout and a source of engagement and satisfaction (the results of the quantitative part of this study are summarised in the Research design sub-section).

Based on the aforementioned literature, in our opinion, it is important to study in more detail the nature and distinctive aspects of burnout—and of its opposite pole, engagement—in a specific, complex, in some ways extreme and still little-studied context, such as that of the reception of illegal immigration.

Our study fits into the JD-R theory perspective of well-being and job distress [15,17,33]. As we have argued, this theoretical framework provides a descriptive model of burnout and its opposite and positive pole, namely engagement [19,21,34]. This model, in turn, defines a set of individual and work dimensions that can be positive (resources) or negative (demands), and which can determine the well-being (engagement) or organisational distress (burnout) of workers [35,36].

We also consider the dichotomous and polarised dimensions that have a positive and negative effect in determining the engagement or burnout of illegal immigration workers. We intend to focus on the understanding of burnout as a syndrome of helping professions with a strong ideal and vocational orientation. For these practitioners, their work takes on a deep and existential meaning; for this reason, they risk not only becoming exhausted, cynical and feeling professionally ineffective [11,12,13], but also seeing their ideals and work values become disillusioned [8,21,24,25,26,27]. This key to interpreting burnout, in the context of illegal immigration management, is in our opinion the most appropriate one, given the considerable number of operators engaged in a voluntary, vocational (and often free) way in this field of aid [32].

Aims: We intend to describe the semantic nature of burnout and engagement in the context of illegal immigration, through an in-depth analysis of the conceptual and thematic cores expressed by the operators involved, distinguishing between the two different operational levels present in this field of aid: rescue/first aid and reception/integration of migrants.

## 2. Materials and Methods

### 2.1. Research Design

The proposed study constitutes a deepening of the research conducted by Nonnis and colleagues [32]. The main objective of the previous study, which was both quantitative and qualitative, was to describe the health and/or burnout conditions of illegal immigration practitioners. From a quantitative point of view, in the previous study, burnout was measured in the overall sample (*n* = 193) with the Link Burnout Questionnaire (LBQ [24,25,26,27]), which measured four dimensions: psychophysical exhaustion, relational deterioration, professional ineffectiveness and disillusion. The qualitative part of the previous study was carried out by means of semi-structured interviews (conducted both face-to-face and by e-mail, *n* = 108). The quantitative results showed that, in general, the sample was in a state of discomfort and distress and that about a quarter (*n* = 51) were at severe risk of burnout (the qualitative results of the previous study are summarised in the Introduction). These findings (and those obtained in the other studies described above) prompted us to further explore the semantic nature of the burnout construct in this specific helping context.

The present study is qualitative and semantic in nature. It was conducted with part of the sample of operators who, in the previous study [32], had answered the LBQ. It was carried out by means of semi-structured, face-to-face, audio-recorded interviews conducted in Italian, which were then transcribed in full using word processing software. The group of participants was identified on the basis of the availability of respondents, using non-probabilistic sampling for representative elements and following a multiple case study approach [37].

A total of 62 face-to-face interviews were carried out, with reference to the two levels of the Italian immigrant reception system (rescue/first aid and SAI). Data were collected between July 2019 and January 2020.

### 2.2. Assessment Instruments

Each participant received an information sheet on the respect of privacy and anonymity (according to Italian law) and a form for collecting socio-demographic data: gender, geographical area of service (north, centre, south or islands), level of reception (rescue/first aid or SAI), being a volunteer or professional, professional qualifications and length of service.

To collect data on the motivations and arguments proposed by the operators regarding their experiences of discomfort (burnout) and organisational well-being (engagement), we applied a face-to-face semi-structured interview that follows the model of burnout proposed by Santinello et al. [24], which has been used in our previous and other studies (e.g., [25,26,27,32]). The model includes the following four dimensions: exhaustion (EXHA) versus energy (ENER), relational deterioration (REL_DET) versus relational involvement (REL_INV), professional inefficacy (PR_INEF) versus professional efficacy (PR_EFF) and disillusion (DISILL) versus trust (TRUST).

The interview consists of eight questions, two for each of the four burnout and engagement dimensions investigated. In each pair of questions (see below), the first one elaborates on the negative and discomforting aspects of the practitioners’ work (burnout), while the second examines the positive and motivating ones (engagement).

Exhaustion versus energy:

“What makes you feel tired in your job?”,

“What makes you feel energetic in your job?”.

Relational deterioration versus relational involvement:

“What makes you feel detached in your relationship with the immigrants?”,

“What makes you feel involved in your relationship with the immigrants?”.

Professional inefficacy versus professional efficacy:

“What makes you feel ineffective in your job?”,

“What makes you feel capable of doing your job?”.

Disillusion versus trust:

“What makes you feel disappointed and disillusioned in your job?”,

“What makes you feel successful in your job?”.

### 2.3. Participants

Sixty-two practitioners working in the Italian reception system for illegal immigrants participated in this study. Specifically, the work included 31 participants working in the rescue and first aid (first-level) structures and 31 in the SAI (second-level) structures. All participants had direct and systematic contact with illegal immigrants, and they agreed to complete a semi-structured interview about their work, assessing the four aforementioned dimensions. Table 1 shows the interviewees’ distribution regarding the geographical area.

Participants expressed a strong fear of being identified and having problems at work, especially if they made critical or negative remarks about the organisation of the reception. This fear was amplified by the small number of workers present in some structures; in other cases, these fears were expressed by the managers and coordinators themselves. For these reasons, in the interviews carried out, the respondents allowed the interviewers to report the level of reception (rescue and first aid or SAI) and the territorial area of service (north, centre, south or islands), but did not allow the reporting of the other socio-demographic information required by the protocol (gender, being a volunteer or professional, qualification and length of service).

### 2.4. Ethical Issues

The research was authorised by the Ethics Committee of the University of Cagliari (approval number 009858 dated 22 May 2020). It was conducted in full compliance with the Ethical Principles of Psychologists and Code of Conduct of the American Psychological Association (APA), integrated into the Associazione Italiana Psicologia (AIP) code of ethics. Furthermore, the study did not address any sensitive topics and was carried out via procedures for informed and consenting adults. Lastly, in accordance with Italian privacy law, the research ensured the anonymity and privacy of all participants.

### 2.5. Methodology and Textual Data Analysis

The semi-structured interviews were transcribed verbatim. The data were examined using the software T-Lab 8.0 [38], devised for the application of a semiautomatic textual analysis (i.e., computer-assisted qualitative data analysis software—CAQDAS [39,40,41]). Specifically, the conceptual mapping of the corpus was applied to connect different portions of text to research hypotheses [42]. This methodology defined the single words as “lexical units” (LUs). These LUs have been identified in the text in two ways, namely in their authentic transcript form or in their headword form, made distinct by the process of “lemmatisation” (a semiautomatic process, reconducting, for example, the word “easier” to its headword form “easy”). Furthermore, subsets of the corpus constituted the “context units” (CUs), in which sentences or paragraphs have been identified and connected with the variables investigated.

To carry out the analyses, the answers were systematised as a dataset and transformed into a .txt file (suitable for T-Lab). The textual corpus was divided in relation to the dimensions investigated, to the polarity of each dimension and to the practitioners’ reception level. The analyses were carried out by allowing the software to organise the corpus in CUs and to apply the process of semiautomatic lemmatisation to the recognised words. The researcher supervised and reviewed these processes, re-examining and revising the unlemmatised words (because they were not in the dictionary of T-Lab). At this point, a customised lemmatisation was applied, observing each unlemmatised word in its sentence context; this control allowed us to clarify the meaning of homograph words (i.e., words that were spelled in the same way, but that potentially had multiple connotations), and the sense of words that needed disambiguation. Subsequently, the lemmatised text was examined, fixing on the occurrences of the use of words in the whole corpus and in specific sub-sections of the text.

Cluster analysis was applied to identify the contextual fields of meanings common among the practitioners interviewed (themes) [43]. The software produced each cluster of words, grouping the words that were frequently used in the same elementary context; these words were ranked based on their chi-squared values (from high to low) [40]. Furthermore, the software allowed us to identify some descriptive variables, evaluated in relation to each thematic cluster. The clusters and the descriptive variables are plotted in a factorial plane, identified on the basis of an analysis of binary lexical correspondences [44,45].

## 3. Results

The original corpus included 17,848 occurrences (tokens), with 3037 types (in the process of analysis, the queries proposed during the interviews were omitted from the corpus). The application of vocabulary customisation was conducted on 2585 lemmas. There were 1683 hapaxes (words used only one time in the corpus). Then, a threshold of four occurrences was established to identify the list of key words required by T-Lab to run the data analysis (*n* = 176 key words).

To highlight specificities and features of the two levels of reception, the corpus was analysed separately in relation to each type of working level. We applied two different thematic analyses of elementary contexts, implemented by the cluster analysis, and carried out by the bisecting K-means algorithm [40,46]. These analyses were applied first to practitioners working at the first level (rescue and first aid) and second to practitioners working at the second level (SAI).

In the following sections, we illustrate the findings that depicted the relationships among clusters, lemmas, CUs and categorical variables.

### 3.1. Burnout and Engagement in the First Level (Rescue and First Aid)

The bottom-up clustering of textual units highlighted themes of interest in relation to each cluster, related to specific “patterns” (i.e., the dimensions of interviews); they were identified by the algorithms that started from the assessment of co-occurrences for each word [40]. Specifically, this cluster analysis emphasised four clusters (Table 2).

Cluster 1, “Realisation and frustration” (68.70% of CUs), includes the most CUs for the first-level structures; it refers to the relevant aspects connected to the work of practitioners, comprising positive and negative aspects. We can identify lemmas like “work”, “to help”, “pleasure”, “our”, “relationship”, “to see”, “to talk”, “years”, “problems” and “to welcome”. These themes are related to both dimensions of disillusion versus trust (negative and positive) and energy (positive) (see Table 2). Some typical sentences are the following: “The passion, the help, the group and the teamwork” and “I like it as a job, it’s stressful because it’s a job that started a few years ago and we’re still unprepared on how to handle these guys, but it’s interesting to have a relationship with them”.

Cluster 2, “Managing difficulties” (19.00% of CUs), includes words such as “home”, “to understand”, “to need”, “to manage”, “to work”, “to succeed”, “to think”, “to carry out”, “to arrive” and “to know”. These lemmas are related to both polarities (negative and positive) of relational deterioration versus relational involvement. Some representative sentences are: “The guys we follow who call their homes saying they are fine and happy in Italy”, and “If you ‘give them a hand they take your arm’, you have to be able to be detached sometimes, to be cold”.

Cluster 3 is defined as “Coping and ineffectiveness” (9.00% of CUs); it concerns the lemmas “to feel”, “difficulty”, “volunteering”, “inadequate”, “immediately”, “experience” and “to cope”. These words imply both negative and positive polarities of professional inefficacy versus professional efficacy. Regarding this theme, the following sentence is representative: “Sometimes I feel inadequate […] because I would like to have more training on the subject of immigration, I don’t know many things, even if I’m working in this field, this in my opinion it’s wrong. We need to be trained”.

Cluster 4 (3.30% of CUs) depicts the aspects related to “Patience and decommitment”, including the lemmas “boy”, “to find”, “people”, “story”, “to clean”, “different” and “to return”. These aspects seem to characterise the text regarding the negative pole of exhaustion, but also the positive pole of relational involvement. This sentence well represents the cluster: “If the guys did a lot of wrong things, for which we have to take action, I would tend to forgive everything, because I know their history and maybe I think: ‘Poor guy, he did that because...’ but this doesn’t help them to grow. You cannot work in this way”.

Figure 1 shows the cluster representation and descriptive variables for the first-level structures (rescue and first aid).

### 3.2. Burnout and Engagement in the Second Level (SAI)

The cluster analysis carried out regarding the second-level structures highlights four thematic clusters, defined on the basis of their characterising lemmas, with the same modalities applied to the first level (Table 3) [40].

Cluster 1 assembles lemmas referring to “Belonging and lack of recognition” (45.00% of CUs). The lemmas strongly used are “work”, “boy”, “to work”, “to feel”, “colleagues”, “to see”, “user” and “pleasure”. The descriptive variables are related to the positive and negative poles of the dimensions exhaustion versus energy and disillusion versus trust. Some typical sentences are: “I do what I was trained to do, I am motivated so I like my team, I like the work, despite the negative aspects and the problems we encounter […]”; “I don’t know exactly what it is that gives me energy, perhaps the fact that with my colleagues we also have moments in which we laugh and joke, or perhaps the fact that I’m in an office in which there’s a kind of calm atmosphere. Of course, we also have moments of panic, especially when there are deadlines or accounts that don’t add up”. Another representative quote is: “I don’t operate in a particularly rewarding working environment; the only positive reinforcement is usually provided by the guests and almost never by the facility. In addition, I would like to ‘take out’ the work we do inside the centre, but I have not found much feedback on this topic and I fear that in this way we do not operate the action of ‘diversity education’ that should be one of the objectives of those who work in social work”.

Cluster 2 refers to 27.00% of CUs; the most significant words are “to achieve”, “to help”, “relationship”, “to need”, “role”, “to speak”, “different” and “practitioner”, all of which point to the aspects of “Help and steadiness”. In this thematic cluster, the text refers to both negative and positive polarities of the dimension relational deterioration versus relational involvement. One relevant quote is: “I enjoy my job, I always have the energy to tackle things, I am satisfied […]. The thing that can make me tired are the shifts, we work all the time, because we are always on call for anything, but I am proud to work for them, it is part of my role. I’m called upon to do certain tasks and I don’t get tired of them”. Another sentence that describes this theme well is: “Many of my colleagues give them appointments and sometimes they don’t show up, so these guys come here exasperated and angry, but I have to protect myself and my role […] we need rigour, these are things I won’t compromise”.

Cluster 3 designates the theme of “Incomprehension and wear” (19.00% of CUs); it includes words such as: “problems”, “language”, “finding”, “difficulty”, “to welcome”, “to carry out”, “to manage”, “to think”, “activity” and “difficult”. The significant descriptive variables are related to the negative polarity of professional inefficacy. Some sentences describe this theme well: “Not being able to handle situations that in another context would be trivial and easily overcome. Language is a big obstacle”; “Some language difficulties due to foreign languages. I would like to be able to express myself better”.

Cluster 4 illustrates the aspects related to “Facing problems” (9.00% of CUs). This cluster contains lemmas like “to understand”, “unexpected”, “patience”, “to ask”, “ability”, “important”, “to lose” and “experience”. This thematic cluster is related to professional efficacy and relational deterioration. It is useful to highlight the following quote: “I’m very organised, unforeseen events do occur, but I say to myself: ‘I’ll sort it out, I don’t need to worry’. I try hard, then if I don’t succeed, I ask my colleagues for help, but I’ve tried. Of course, if I find an unforeseen problem, I’m disconcerted, but I do try, and in this job you need a great capacity for problem solving. I have learned to negotiate with myself. When you have to deal with a phenomenon such as immigration, which […] is very badly managed in Italy on all fronts, everyone ‘makes fire with the wood they have’. You have to think about the resources that each of us has not to solve the problem, but to face it”.

Figure 2 shows the cluster representation and descriptive variables for the second-level structures (SAI).

## 4. Discussion

The themes proposed by the operators and volunteers of the Italian system of the reception of illegal immigrants are consistent with the experiences and narratives described in the literature on burnout, especially in the contexts of help [10,11,21,23,24,32]. However, it is possible to identify some differences between the two levels of the reception system.

The first level (rescue and first aid) is organised for interventions often characterised by urgency and emergency, risk and time pressure; these interventions often concern the rescue of migrants who are experiencing serious difficulties or whose lives are at risk. This level also includes the management of the most important and urgent health, psychological, legal and humanitarian problems of migrants. It is designed to host migrants for a relatively limited period (a few weeks or a few months at most); time to improve their psychophysical condition and to establish whether they are entitled to any kind of protection. In this scenario, burnout and engagement of practitioners and volunteers emerges in their commitment and satisfaction with the success of their work and frustration with the failure of their efforts (Realisation and frustration). It also reveals their commitment to managing difficulties and contingencies (Managing difficulties), organisational and training deficits (Coping and ineffectiveness), and their tolerance (but also discouragement) in the face of migrants’ suffering and discomfort (Patience and decommitment). These results, which confirm the findings of previous studies (e.g., [28,31]), are well represented by the following sentences from interviews.

“In operations you know when you start, but not when you finish”.

“The work I do [...], I do because I enjoy helping others. This work gives me joy. [...] It is we who have difficulty in helping them because we encounter too many obstacles in our work”.

“The thing I like is the relationship with immigrants, with those who really need international protection and are fleeing from war [...]. Don’t think that everything is so perfect here. [...] Anyway, I studied to do this job, this is what gives me energy”.

“[...] There is bad faith, some of them are lying and it is obvious that they are here without justification so why should they have the same treatment as those who really need to be welcomed? When I realise that there are kids who are not fleeing from war and poverty, I freeze up, it’s as if I lack empathy and solidarity on my part. I define my users as difficult, I work above all with minors and believe me that it is not easy to manage them because they arrive here with an arrogance, a lack of respect and an experience in their country of origin that Italian minors can only dream of. They are already adults in their own right”.

“Many of the boys [...] escape from war or poverty and I get on well with them because they are sensitive boys who need support, many are my son’s age and I feel like a father to them. With these children, I am empathetic, I care about them [...]”.

“There is a difficulty in meeting cultures, they make me angry. I rarely give in and I try to make them understand that things have to be done according to our rules. [...] I often realise that I am cold to the rude ones [...]. They tend to get angry straight away, [...] I make them see reason by saying: ‘See, what are you like? You’re not going anywhere if you’re like this’ and when they see that you’re giving them useful advice then they listen to you and calm down; even if in the meantime they have insulted you or said bad words to you because they feel they are being mocked and attacked”.

“[...] In this job, dialogue is the key role, you have to be charismatic, you have to know how to do it and educate these children, especially minors. [....] I feel I can do it, they listen to me […] and often ask me for advice. To do this job we mustn’t be racist, during the interview the directors test us to understand our attitude towards them; so it’s a special job because it’s not for everyone. I feel that I am capable of doing this job”.

The second reception level (SAI) is intended for immigrants who are waiting for or have been granted refugee status. It therefore implies longer residence times for migrants (on the order of months and in some cases years), has objectives of social inclusion and integration and implies a complex process of interaction with agencies in the territory (e.g., educational, labour and administrative). Again, burnout and engagement of staff and volunteers are shaped by the nature of the work and the objectives of this level of care. In fact, aspects related to the relationship with refugees and asylum seekers emerge: the sense of belonging to the team of operators, the sense of community in the centres and the lack of social recognition of their commitment to the integration of users (Belonging and lack of recognition); listening, helping and steadiness in managing the rules of coexistence (Help and steadiness); greater mutual knowledge and the possible wear and tear of interpersonal relations (Incomprehension and wear); and the difficulty of and commitment to managing the complex processes of collaboration with the territory (Facing problems). These findings, which confirm some of the results of previous studies (e.g., [29,30,32]), are well represented by the following sentences.

“[...] I rarely look at the clock to see how long it is before I finish work [...], Yes, I get tired [...] I am forced to make considerable and prolonged efforts to manage what I call the ‘non-problems’, i.e., one of those activities which in its banality and ease is not managed correctly and generates a difficulty because it is done badly or neglected”.

“I’m very patient and working in a team [...] it’s difficult to get everyone to agree on certain decisions, but I generally manage to have charisma among my colleagues and I plan everything, I try to fit everything in, leaving me room for the unexpected. Unfortunately, unforeseen events are common in our work, but I can manage them, [...] experience has taught me to understand when I can manage an unforeseen event on my own or when external intervention is required [...]. I am often overloaded with requests and the mobile phone [...] becomes my worst enemy because [...] we are always waiting for some confirmation for internships or projects, confirmations for authorisations, so when I miss an important call I get angry because calling back [...] the municipality or the prefecture is not easy, they also have work to do, so when they consider you, you have to take the opportunity immediately”.

“[...] The thing that makes me lose my patience and affects my availability to young people is the way they behave, they often demand a job without learning the language or without doing some training to learn a trade, so I become cynical and I treat them badly to try to stimulate them and make them react. [...] I must also be provocative, because otherwise they would be [...] waiting for ‘divine help’ [...] but at the same time they get angry because they don’t think we’re doing enough for them [...] you have to try to [...] make them understand that it’s up to them [...]”.

“In my sector there is a risk of falling into a ‘delirium of salvation’, you can’t save people, you can help them to work out their problems or needs, but there is no immediate and obvious solution and sometimes you may feel not so much inadequate as disappointed by your work, but it’s part of the job, you have to work on this aspect too; I get very tired of this job because often things don’t go as you want them to, but […] you either love it or you don’t”.

Based on these results, it is possible to propose an overall interpretation of the relationship between engagement and burnout in the context of illegal immigration management. As previously argued, the JD-R theory [15,17,33] conceives of burnout and engagement as polar opposites. It identifies job demands that can provoke practitioners’ burnout and job resources that can protect them from it and foster their engagement. For example, Schaufeli et al. [35] identified social support, autonomy, opportunities to learn and feedback as resources able to nurture engagement, and workload, emotional demands and work–home interference as the main job demands able to determine employee burnout. Hu and colleagues [36] identified, in addition to the above dimensions, participation in decision making and job control as job resources, and interpersonal conflict as job demands.

By contrast, our results show that in the vast majority of the dimensions considered—three out of four for both levels of reception—the same dimensions in which the engagement of the operators is declined (energy, relational involvement, professional efficacy and trust) are able to lead them into a condition of burnout (with experiences, on the opposite side, of exhaustion, relational deterioration, professional inefficacy and disillusion). In fact, regarding the first level (rescue and first aid), the realisation and frustration of the operators can be traced back to the positive and negative dimensions of disillusion versus trust; having to manage difficulties can be traced back to the positive and negative dimensions of relational deterioration versus relational involvement; and their coping and ineffectiveness are associated with the positive and negative dimensions of professional inefficacy versus professional efficacy. Considering the second level (SAI), the experiences of belonging and lack of recognition of the practitioners are associated with the positive and negative dimensions exhaustion versus energy and disillusion versus trust, and their experiences of help and steadiness are related to the positive and negative dimensions of relational deterioration versus relational involvement. These results are in line with the literature—both historical and more recent [8,9,21,24,25,26,27]—that frames burnout within the context of professions strongly characterised by motivation and the value of helping others.

Our study, given the paucity of literature on the subject, deepens knowledge on the nature of burnout in the under-researched context of illegal immigration and could enable future research in this area. One aim could be to confirm the results of this study by using objective measures (e.g., indicators of health or work stress, or relating to organisational performance), in addition to the subjective and qualitative indicators adopted to date. Another one, of a more applicative and certainly useful nature, could concern the implementation and evaluation of the effectiveness of actions and interventions to nurture and enhance the personal resources of illegal immigration practitioners, rendering them able to maintain an adequate level of engagement and thus prevent burnout.

This study has some limitations. The first is that we used convenience sampling (the availability of participants), without the aid of statistical control procedures for selecting cases. The second limitation lies especially in the subjective nature of the data collected. Given the delicacy and criticality of the survey context, it was not possible to obtain objective data (e.g., requests for sick leave, staff data, etc.) that are indicative of conditions of work-related stress or burnout. The third limitation is the impossibility of distinguishing the study results on the basis of certain socio-demographic dimensions (gender, job role, length of service, being a volunteer or professional), given the unwillingness of the participants to communicate their personal data to the surveyors.

## 5. Conclusions

Our study deepens knowledge on the nature of burnout in the under-researched context of illegal immigration. Moreover, from in the answers given by the interviewees, we found that the same job resources that allow workers to engage and feel effective in helping illegal immigrants’ suffering and problems can, on the contrary, become job demands and determine their burnout.

A plausible explanation for these results can be found in the fact that people whose jobs involve in such an overwhelming phenomenon as illegal immigration are driven by a motivation, vocation and values that go beyond “simply” working in the service sector. Due to this strong idealistic motivation, overcommitment to helping migrants—especially in the face of the great suffering they bear and the logistical and organisational difficulties and shortcomings that often characterise reception facilities—could be the main cause of their burnout.

This possible interpretation of our results, in order to be confirmed, needs to be deepened and evaluated by further research in the specific field of illegal immigration and in other fields of the helping professions where the motivation and value of help play a central role, both with respect to engagement and to its “dark side”, namely burnout.

This study emphasises the importance of taking into account the “two souls” of burnout within the service sector: the one related to services to others in general and the one unique to the helping professions. With reference to the theoretical framework of the JD-R model of burnout, these two perspectives of the possible causes of the syndrome are in our opinion not antithetical. In fact, the dimensions of burnout that in our study are personal job resources (energy, relational involvement, professional efficacy and trust) can be considered valuable sources of engagement and motivation able to feed the management system of irregular immigration (and perhaps any system of the management of helping others). However, these sources should not be abused and all necessary precautions should be taken so that they do not turn into their opposite, thus becoming an overload of demand (with experiences of exhaustion, relational deterioration, professional inefficacy and disillusion), and therefore burnout.

These considerations call into question the programmes and actions that institutional bodies should deploy in order to implement adequate forms of maintenance, monitoring, refreshment and enhancement of these motivational resources. Unfortunately, this study and the others we have mentioned show that governments and public institutions are often more oriented towards exploiting and using these motivational resources, rather than preserving and nurturing them. All of this risks leading to a condition of perpetual emergency and breathlessness that is, on the whole, incompatible with the phenomenon of illegal immigration, which will probably characterise the coming decades and requires an adequate and structured systemic response (at both the national and European levels), that can count on human resources that are adequately motivated and trained for this purpose and are in a state of well-being at work.

## Figures and Tables

**Figure 1 ijerph-18-03726-f001:**
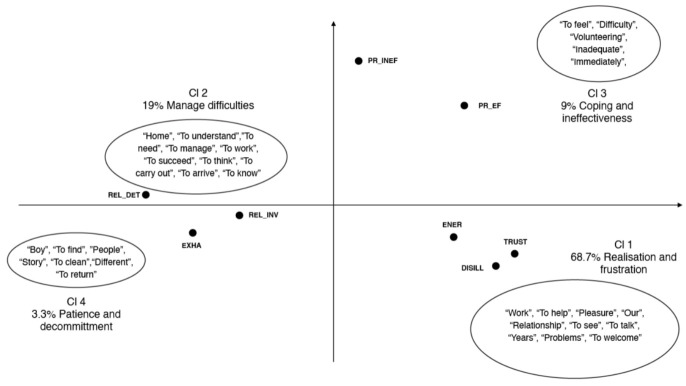
Representation of the clusters and descriptive variables on the factorial plane for the first-level structures. The descriptive variables are: Exhaustion (EXHA), Energy (ENER), Relational deterioration (REL_DET), Relational involvement (REL_INV), Professional inefficacy (PR_INEF), Professional efficacy (PR_EFF), Disillusion (DISILL), Trust (TRUST).

**Figure 2 ijerph-18-03726-f002:**
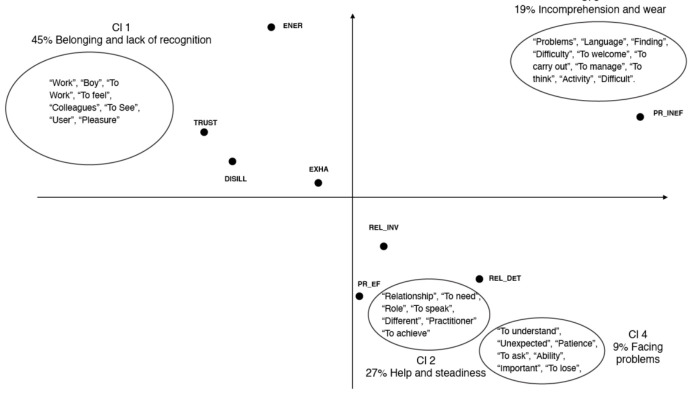
Representation of the clusters and descriptive variables on the factorial plane for the second-level structures. The descriptive variables are: Exhaustion (EXHA), Energy (ENER), Relational deterioration (REL_DET), Relational involvement (REL_INV), Professional inefficacy (PR_INEF), Professional efficacy (PR_EFF), Disillusion (DISILL), Trust (TRUST).

**Table 1 ijerph-18-03726-t001:** Distribution of the interviews with respect to the level of reception and territorial area.

Reception Level	Northern Italy	Central Italy	Southern Italy	Islands	Total
Rescue and first aid (first level)	- ^1^	3	12	16	31
SAI ^2^ (second level)	9	6	6	10	31

^1^ There are no rescue facilities in the north of Italy. ^2^ Reception and Integration System (in Italian: Sistema di Accoglienza e Integrazione, SAI).

**Table 2 ijerph-18-03726-t002:** Results of cluster analysis on first-level structures.

Cluster	% of Context Units (CUs) in the Cluster	Label	Principal Lemmas Ordered by Highest to Lowest Frequency	Descriptive Variables Ordered by Decreasing Chi-Squared Value (in Brackets)
1	68.70	Realisation and frustration	Lavoro—Work (129) Aiutare—To help (47) Piacere—Pleasure (44) Nostro—Our (24) Rapporto—Relationship (23) Vedere—To see (18) Parlare—To talk (18) Anni—Years (17) Problemi—Problems (16) Accogliere—To welcome (16)	Trust (42.907) Energy (35.126) Disillusion (32.317)
2	19.00	Managing difficulties	Casa—Home (36) Capire—To understand (30) Aver bisogno—To need (29) Gestire—To manage (26) Lavorare—To work (25) Riuscire—To succeed (25) Pensare—To think (22) Svolgere—To carry out (17) Arrivare—To arrive (16) Conoscere—To know (15)	Relational deterioration (41.172) Relational involvement (22.833)
3	9.00	Coping and ineffectiveness	Sentire—To feel (63) Difficoltà—Difficulty (15) Volontariato—Volunteering (10) Inadeguato—Inadequate (9) Subito—Immediately (8) Esperienza—Experience (8) Affrontare—To cope (8)	Professional inefficacy (72.911) Professional efficacy (42.020)
4	3.30	Patience and decommitment	Ragazzo—Boy (75) Trovare—To find (28) Persone—People (20) Storia—Story (15) Pulire—To clean (13) Diverso—Different (12) Tornare—To return (11)	Exhaustion (37.210) Relational involvement (5.268)

**Table 3 ijerph-18-03726-t003:** Results of cluster analysis on second-level structures.

Cluster	% of Context Units (CUs) in the Cluster	Label	Principal Lemmas Ordered from Highest to Lowest Frequency	Descriptive Variables Ordered by Decreasing Chi-Squared Value (in Brackets)
1	45.00	Belonging and lack of recognition	Lavoro—Work (107) Ragazzo—Boy (48) Lavorare—To Work (30) Sentire—To feel (29) Colleghi—Colleagues (26) Vedere—To See (24) Utente—User (17) Piacere—Pleasure (16)	Energy (24.821) Trust (18.639) Disillusion (10.952) Exhaustion (8.195)
2	27.00	Help and steadiness	Riuscire—To achieve (37) Aiutare—To help (25) Rapporto—Relationship (24) Aver bisogno—To need (18) Ruolo—Role (17) Parlare—To speak (15) Diverso—Different (13) Operatore—Practitioner (12)	Relational involvement (76.846) Relational deterioration (8.156)
3	19.00	Incomprehension and wear	Problemi—Problems (21) Lingua—Language (20) Trovare—Finding (18) Difficoltà—Difficulty (18) Accogliere—To welcome (16) Svolgere—To carry out (15) Gestire—To manage (14) Pensare—To think (14) Attività—Activity (11) Difficile—Difficult (10)	Professional inefficacy (103.55)
4	9.00	Facing problems	Capire—To understand (15) Imprevisto—Unexpected (13) Pazienza—Patience (13) Chiedere—To ask (10) Capacità—Ability (8) Importante—Important (8) Perdere—To lose (8) Esperienza—Experience (8)	Professional efficacy (80.845) Relational deterioration (13.546)

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
