# Peer review of "Burnout and Engagement Dimensions in the Reception System of Illegal Immigration in the Mediterranean Sea. A Qualitative Study on a Sample of Italian Practitioners"

_ijerph, 2021, doi:10.3390/ijerph18073726_

Round 1

Reviewer 1 Report

The paper addresses a critical challenge  regarding burnout and engagement phenomena among practitioners working in the illegal immigration reception system, seeking for means that can improve an organization’s capability to deal with them. The authors claim for an understanding of burnout as a syndrome of helping professions with a strong ideal and vocational orientation. I think that this ossimoric evidence could be better anlyzed in the discussion, since it opens relevant questions about the assumptions of the JD-R Theory: the same dimensions can be both resources for engagement and demands linked to the burnout, dealing with a contradictory stance at stake that has to be assumed and faced to.

I suggest the authors to add some excerpts of the textual utterances related to the written transcriptiosn of the interviews, integrating tha tables of the T-lab cluster, and highlighting in this way how the practitioners involved seek to manage this contradiction.

I sugest a very little minor revision in order to better highlight in the discussion the discoursive manifestations (a fiew utterances taken from the transcriprions) of the meanings practitioners give to their work experience.

Author Response

Dear Reviewer,

thank you for your suggestions and corrections. The attached tables describe our changes to the article, based on all your comments.

Best Regards

The Authors

Reviewer 2 Report

INTRODUCTION

  • The introduction should not be to have subsections and is very long. The four subsections must be joined into one.

MATERIALS AND METHODS

Research design:

  • Why were some interviews face-to-face and others by e-mail? This is an interviewer bias.
  • The authors must include the response rate of the participants in the study.

2.2. Assessment instruments

  • The authors speak of workers or volunteers. Volunteer staff have different motivations for doing their job. It is difficult to have burnout. This is a selection bias.
  • You cannot see if a person has burnout by means of a semi-structured interview. That is wrong. There are many scales to identify people with burnout.

CONCLUSIONS

The manuscript has not deepened into the study of burnout. The conclusions are wrong.

REFERENCES

  • Much bibliography is not papers published in scientific journals (43,2%). They are web pages and books. There is an updated bibliography of original and meta-analytic articles that should be cited, among others.
  • Many bibliographies are obsolete and some citations are incomplete. The bibliographic citations used are more than 5 years old (32%). The authors must update and arrange the bibliography.
  • Some references that have errors. The authors should review this section.

Author Response

Dear Reviewer,

thank you for your suggestions and corrections. The attached table describe our changes to the article, based on all your comments.

Best Regards

The Authors

Reviewer 3 Report

The manuscript presented is very interesting and highly relevant. It is worth paying attention to the emotional aspects of the professionals who are part of the care system for migrants in Italy.

On the other hand, I must congratulate the authors for using a qualitative methodology and for approaching a topic that usually occupies very few pages in current research.

The following are some recommendations for improving the manuscript presented:
- It would be interesting to adapt the abstract to a classic format in which the objective is clearly reflected. The last two sentences, moreover, are not very specific. It would be worthwhile to reshape the abstract to make it easier to read.
- The first sentence of the introduction needs to be supported by bibliographical references.
- The objective should appear at the end of the introduction.
- On the other hand, it would be interesting to make the objective more precise. I understand that what the authors are looking for is to describe the burnout of the workers, not the context. Is this the case?
- In the discussion, it would be interesting to include possible lines of future research based on the work carried out. 

Author Response

(The authors gave the same response as above.)

Reviewer 4 Report

The issue of burnout of operators working in the field of reception of illegal immigration is certainly a topic of great importance and not yet sufficiently explored in the scientific literature.

However, I must point out to the authors a serious incompleteness in their manuscript. The bibliographical references do not follow the rules of the journal and the numbers included in the text are not matched in the list of citations . This makes it very difficult for the reader to perform a review of the soundness and congruence of the theoretical framework.

Apart from the aforementioned criticality, the research is described with clarity and rigor, the authors provide an exhaustive framework in which to frame their study while also highlighting the gaps with respect to the current knowledge on burnout of operators and volunteers. 

From the methodological point of view the descriptive model is rightly linked to the theory and explained in detail in every step.
The results are evident and their discussion well highlights the link between contribution to new knowledge and gaps highlighted in the literature review.

In order to enhance the value of their study, I suggest that the authors better emphasize in their conclusions the theoretical contribution of their study and the implications that the results could have on migration governance policies. 

Author Response

(The authors gave the same response as above.)

Round 2

Reviewer 2 Report

Dear authors,

You say that this investigation is the continuation of another one already published. The data of the manuscript is insufficient to be able to replicate it.

If they had burnout in the previous study it does not mean they have it today. In addition, with the information that you expose it is not possible to know if your “Conclusions” are correct.

In addition, the following is important:

MATERIALS AND METHODS

  • The authors say that this research is a continuation of another study. An investigation must be able to be replicated and in this case it cannot be.
  • There is no response rate because no sampling has been done. What has been done in another study does not determine what is done in this research. This is wrong. Can't replicate.
  • Missing operator data can be confusing if not identified. This is wrong and is a major limitation. We are in the same case, the study cannot be replicated.

Author Response

Dear Reviewer 2,

thank you for your suggestions and corrections. The attached table describe our changes to the article, based on all your comments.

Best Regards

The Authors

Reviewer 3 Report

The modifications made by the authors satisfy the requirements of this reviewer.

Author Response

Dear Reviewer 3,

thank you for your suggestions and corrections. The attached table describe our changes to the article, based on your comment.

Best Regards

The Authors
